# Novel insights into neuropathy: The impact of prolonged hyperglycemia on long non-coding RNA expression

Kamila Zglejc-Waszak[1]☉*, Jan Paweł Jastrzebski[2,3]☉, Joanna Wojtkiewicz[4], Zenon Pidsudko[1], Judyta Karolina Juranek[4]

**1** Department of Anatomy and Histology, School of Medicine, Collegium Medicum, University of Warmia and Mazury in Olsztyn, Olsztyn, Poland, **2** Department of Plant Physiology, Genetics and Biotechnology, Faculty of Biology and Biotechnology, University of Warmia and Mazury in Olsztyn, Olsztyn, Poland, **3** Bioinformatics Core Facility, Faculty of Biology and Biotechnology, University of Warmia and Mazury in Olsztyn, Olsztyn, Poland, **4** Department of Human Physiology and Pathophysiology, School of Medicine, Collegium Medicum, University of Warmia and Mazury in Olsztyn, Olsztyn, Poland

☉ These authors contributed equally to this work.
* kamilazglejc@gmail.com, kamila.zglejc@uwm.edu.pl

## Abstract

Multiple evidence suggests that type 1 diabetes triggers perturbations in the nervous system both in human patients as well as in animal models of the disease. These perturbations are likely controlled by the expression of long non-coding RNAs (lncRNAs) and are present both in peripheral and central nervous system. To dissect the role of lncRNAs in diabetes-affected nervous system malfunctions, we conducted a comparative analysis of spinal cord transcriptome profiles between long-term (six months of duration) diabetic versus non-diabetic mice. The analysis of RNA sequencing data revealed that of 277 unique differentially expressed transcripts, 201 were up-regulated and 76 were down-regulated in the diabetic lumbar spinal cord. We also observed elevated expression of *Snhg15* lncRNA in diabetic spinal cord. The in-depth data analysis revealed differential expression of lncRNAs involved in the PI3K-Akt signaling pathway (KEGG: mmu04151) as well as substantial differences in several biological processes such as developmental process, cell communication, anatomical structure development and multicellular organismal process. Our analysis verified the role of lncRNAs in mouse spinal cord during the progression of type 1 diabetes and confirmed molecular alternations in the spinal cord occurring in the course of diabetic neuropathy.

## Introduction

Numerous studies have shown that type 1 diabetes (T1D) is the main cause of peripheral nervous system perturbations [1–6]. Studies indicated that patients with diabetes have also malfunctions in the central nervous system [7–11]. Our previous RNA-seq

**Data availability statement:** All relevant data are within the manuscript and supporting information. RNA-seq data have been deposited in the ArrayExpress database at EMBL-EBI (www.ebi.ac.uk/arrayexpress) under accession number E-MTAB 12252.

**Funding:** This work was supported by the National Science Centre, Poland; Grant no. UMO-2018/30/E/NZ5/00458. The publication fee was funded by the School of Medicine, *Collegium Medicum*, University of Warmia and Mazury in Olsztyn, Poland; Grant no. 61.610.100-110. The funders were not involved in the design or analysis of this research, the preparation of the manuscript, or the decision to publish.

**Competing interests:** The authors declare no conflict of interest.

studies indicated that long-term T1D may alter the expression of genes in the mouse spinal cord [2]. We observed that the expression of 248 differential genes increased and 137 decreased in lumbar spinal cord harvested six months after T1D induction in mice [2]. Moreover, our data demonstrated that changes in the spinal cord occur simultaneously with molecular changes in the sciatic nerve during T1D [2]. However, the cause of molecular changes in the nervous system during T1D is still unknown.

Nevertheless, studies indicated that long non-coding RNAs (lncRNAs) may play a key role in neurological complications of diabetes [12], likely affecting the receptor for advance glycation end products-diaphanous related fromins 1 (RAGE-Diaph1) signaling pathways in T1D nervous system. Our previous studies have shown that the interaction between RAGE and Diaph1 is essential in neurological complications [2,6,13,14].

Results suggest that lncRNAs may influence gene expression in the mouse lumbar spinal cord during long-term T1D [2,3,15–17]. Therefore, the aim of the present study was to investigate the expression of lncRNAs and the relationships between target genes in the lumbar spinal cord harvested from mice with T1D. We observed that lncRNAs may have an impact on the PI3K-Akt signaling pathway in the lumbar spinal cord and thus contribute to the progression of T1D.

## Materials and methods

### Animals and tissues

The study was approved by the Local Ethics Committee of Experiments on Animals in Olsztyn (Poland; decision no. 57/2019).

Male mice (C57BL/6) that had T1D for 6 months were obtained as previously described [3] and the presence of peripheral neuropathy was confirmed [3]. Briefly, at eight weeks of age mice were randomly divided into two experimental groups (n = 5 per group) and treated with: streptozotocin (STZ) or vehicle (PBS, phosphate buffer saline). T1D was induced by intraperitoneal injection of 50 mg/kg STZ diluted in PBS for five consecutive days. Simultaneously, our control group received the same volume of PBS throughout treatment period [2]. Among T1D mice, animals with a glucose concentration of 13 mmol/L (260 mg/dL) were selected for further experiments. Blood glucose measurements as well as the confirmation of diabetic neuropathy were previously described [2]. Diabetic peripheral neuropathy was confirmed by measuring nerve conduction and determining the ultrastructure of the sciatic nerve (data included in the previous manuscript [2]). As described previously [2], all mice were anesthetized using ketamine (300 mg/kg) and xylazine (30 mg/kg) mixture to minimize mouse suffering. Mice were humanely euthanized by cervical dislocation. Following collection, lumbar spinal cords were collected on ice, immediately frozen in liquid nitrogen and stored at −80 °C until further analyses. All tissues were sampled during our previous study [3], therefore we limited the number of animals used in accordance with the 3R principle [18].

### RNA isolation

Total RNA was isolated the use of a RNeasy Plus Mini Kit (Qiagen, Hilden, Germany). Genomic DNA (gDNA) contamination was effectively removed using a

special gDNA eliminator spin column. The purified RNA was ready to use for further analyses, such as: Next Generation Sequencing and quantitative PCR (qPCR).

## Next generation sequencing

The sequencing reactions were performed as described in Zglejc-Waszak and co-workers [2]. Briefly, we have used the NovaSeq 6000 platform (IlluminaR, USA) to generate 2 × 150 bp reads. We have obtained 40 million readings per sample. Next, sequencing data was converted into raw data for the *in-silico* analysis of lncRNAs [18–19].

## Bioinformatic analysis of lncRNAs

We performed *in silico* preprocessing including both quality control using FastQC software version 0.11.7 (Bioinformatics Group at the Babraham Institute, Cambridge, UK; www.bioinformatics.babraham.ac.uk) and trimming of low quality reads (Phred cut score ≤ 20; reads length = 90 nucleotides, default Illumina adapters removing) using Trimmomatic version 0.39 [20]. Mapping process was performed using STAR, v. 2.7.10a and StringTie, v. 2.1.7. based on the reference mouse genome GRCm39 (Genome Reference Consortium Mouse Reference 39), INSDC Assembly GCA_000001635.9 (ENSEMBL release 102) and annotation version 107 [21–22].

The lncRNA identification process followed the previously described procedure [19] applying the lncRna library [23] for the validation and the setting of a customized set of procedures with the most optimal configuration for these studies. We implemented the following methods to identify coding potential: "CPC2", "PLEK", "FEELnc", "CPAT", "CNCI", "LncFinder" [24–26]. Finally (using the lncRna library), we selected two tools for the combination of coding potential analysis methods: CPAT and CNCI as the most optimal solution for this particular study (Fig 1).

Thanks to the very favorable ratio between correct predictions and incorrect ones, and to the high-quality annotation of the reference genome, 1-exon transcripts were not filtered out Therefore, the final pool of predicted new lncRNAs also includes 1-exon transcripts.

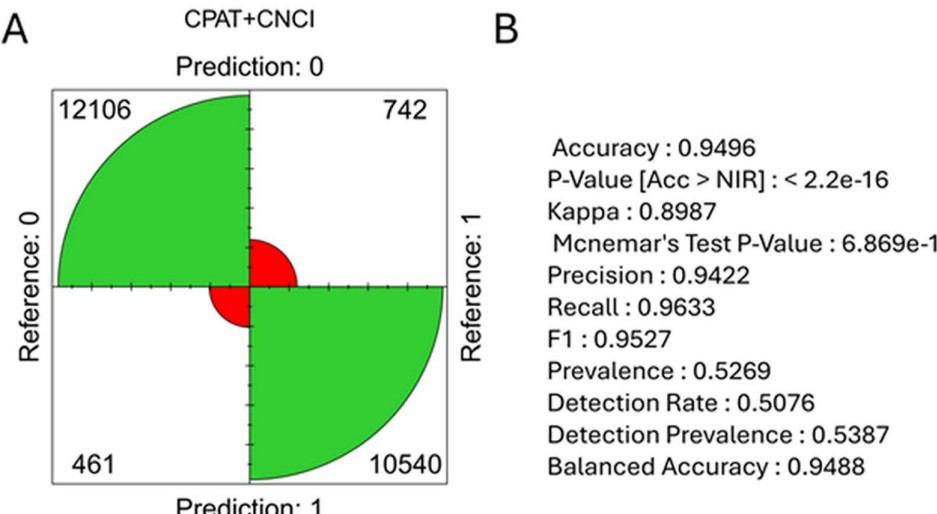

**Fig 1. FourPlot. The plot of confusion matrix (A) generated for the combination of the coding potential prediction results of the two selected tools: CPAT and CNCI.** Green fields represent truly and red – falsely predicted features. Statistics confirming the quality of the prediction by the selected combination of methods (such as accuracy, P-Value or precision) are shown in part (B) of the figure.

The STRING v. 12 database was used to created interaction networks (https://string-db.org/). REVIGO server was used to summarize and visualize lists of Gene Ontology (GO) terms [25–26]. David KEGG database was used to visualize biological pathways [26].

## qPCR analysis of lncRNA

RNA was transcribed (QuantiNova Reverse Transcription Kit (Qiagen, Hilden, Germany) to cDNA according to the manufacturer's instructions [14]. *Snhg15* expression was tested in duplicates using a qPCR with SYBR® Green PCR Master Mix (Qiagen, Hilden, Germany; [14]). Fold change was calculated using ΔΔCt method and normalized using reference gene (Table 1; [28]). The specificity of qPCR was confirmed by agarose gel electrophoresis.

## Statistical analysis

Results of qPCR were presented as the mean ± SEM. Analyses with P-values ≤ 0.05 were considered statistically significant. Statistical analyses were performed using Statistica 13.3 (StatSoft Inc., Tulsa, OK, USA). Statistical graphs were performed using GraphPad Prism 9.1.0. (CA, USA) as well as Office package.

# Results

## Description of lncRNAs

We obtained 277 differentially expressed transcripts; 201 were up-regulated and 76 were down-regulated in the T1D lumbar spinal cord (Fig 2).

Of these, 181 transcripts were products of lncRNA biotype genes, 78 protein coding and 18 as pseudogenes (Fig 3A). We identified 277 such transcripts, which are lncRNAs, but at the gene level it is as in Fig 3A, and at the transcript level it is as in Fig 3B). A table containing a complete list of lncRNAs that were differentially expressed in the T1D lumbar spinal cord in relation to control is presented in the supplementary data (S1 Table).

## Interaction of lncRNAs with protein

Analysis of direct lncRNA-protein interactions revealed strong associations with three proteins, A disintegrin-like and metallopeptidase (reprolysin type) with thrombospondin type 1 motifs 3 (Adamts3), A disintegrin and metalloproteinase with thrombospondin motif 18 (Adamts18), SCO-spondin (Sspo). Sspo is involved in the modulation of neuronal aggregation. It may be involved in developmental events during the formation of the central nervous system. Moreover, Sspo protein belongs to the thrombospondin family. *In silico* analysis revealed that these proteins participate in Gene Ontology-term (GO-term) associated with Cellular Component – Extracellular matrix (GO:0031012); limited to *Mus musculus* (Fig 4). The most enriched biological pathway was O-glycosylation of TSR domain-containing proteins (MMU-5173214). The mentioned proteins do not form direct interactions, but they create a network of interactions with other proteins (Fig 4). We observed that in the O-glycosylation of TSR domain-containing proteins pathway, Adamts3, Adamts18, Sspo interact with Beta-1,3-glucosyltransferase (B3glct) and GDP-fucose protein O-fucosyltransferase 2 (Pofut2; Fig 5).

**Table 1. Primers used for *qPCR*.**

| Symbol (official) | Transcript | Primers sequences | Accession number | Amplicon length | Reference |
|---|---|---|---|---|---|
| *Snhg15* | *Snhg15–201* | F: TAGCACTTCAGAGACCATCAG R: TGTCTTCAGACACACCAGAG | NR_045893 | 214 nt | – |
| *18S rRNA* | – | F: GGGAGCCTGAGAAACGGC R: GGGTCGGGAGTGGGTAATTT | NR_003278.3 | 68 nt | [27] |

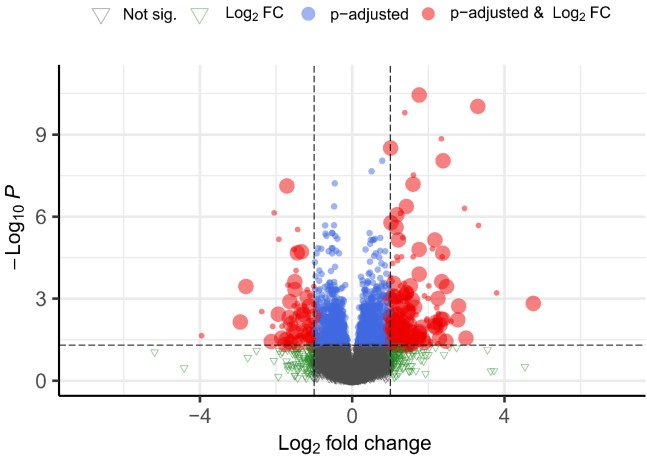

**Fig 2. Volcano plot showing log2fold change plotted against normalized p-values.** Each point (dot, circle and triangle) represent gene expression value. Gray triangle – no significance. Green triangle – Log2FC. Blue circles are genes that meet the significance threshold, but the expression difference is less than 2 (2 x fold change=1 log2 fold change). Red circles indicate genes that meet both conditions, so they are classified as genes with significantly variable expressions, while circles with large diameters indicate lncRNAs with significantly different expressions (DELs). Logarithm of fold change on the X-axis and statistical significance of the measurement (negative logarithm of p-adjusted on the Y-axis.

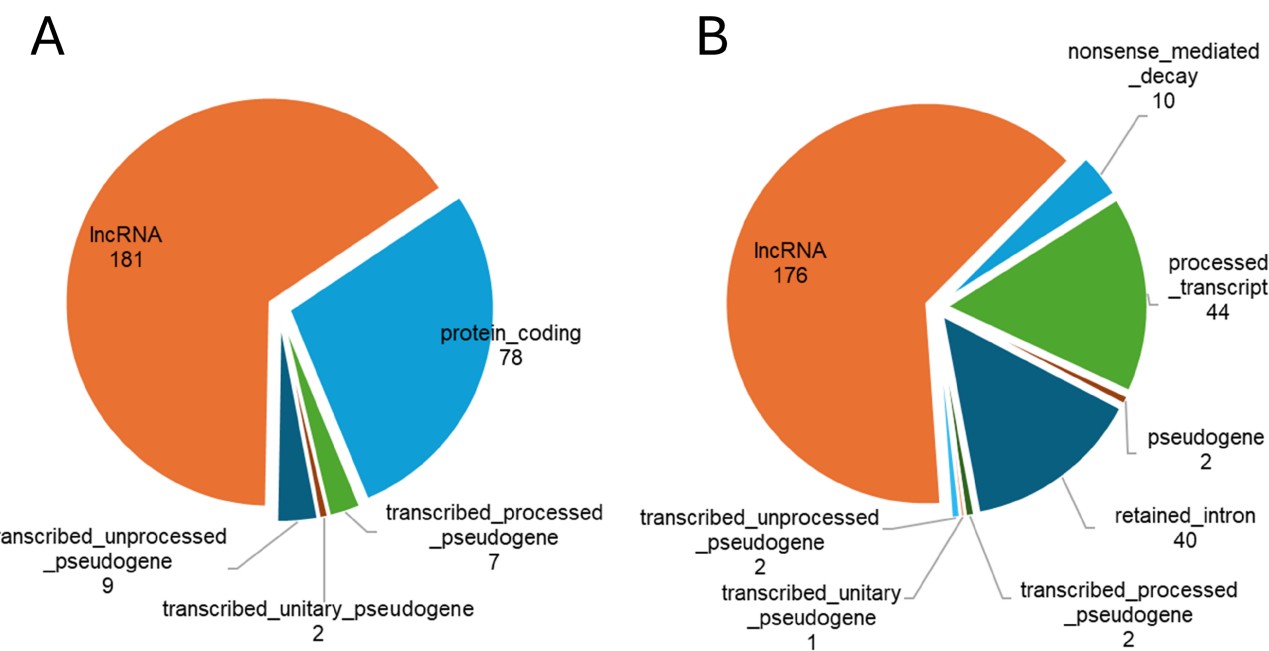

**Fig 3. Description of differentially regulated lncRNAs.** Overall, 277 transcripts were expressed in mouse lumbar spinal cord, while 201 were up-regulated and 76 were down-regulated in T1D mice. Transcripts were divided into five categories based on biotype: protein coding (66 were up- and 13 were down-regulated in T1D mice), lncRNA (117 were up- and 63 were down-regulated), transcribed unprocessed pseudogene (seven were up-regulated in T1D mice), transcribed processed pseudogene (nine were up-regulated in T1D mice) and transcribed unitary pseudogene (two were down-regulated in T1D mice).

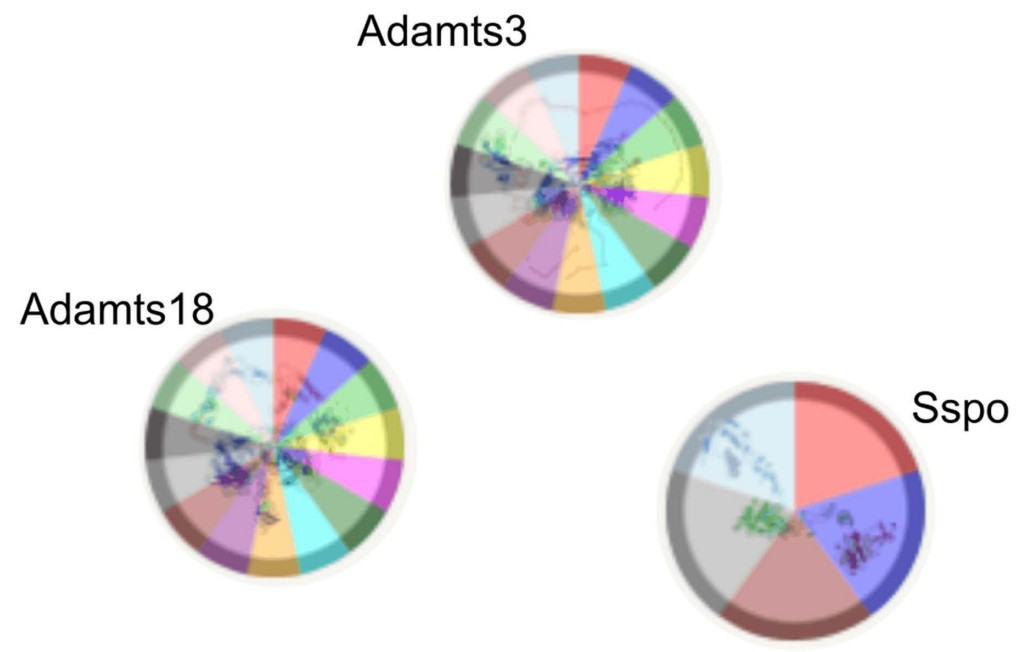

Adamts3

Adamts18

Sspo

**Functional enrichments in your network**

*explain columns*

**Cellular Component (Gene Ontology)**

| GO-term | description | count in network | strength | false discovery rate | |
|---------|-------------|------------------|----------|---------------------|---|
| GO:0031012 | Extracellular matrix | 3 of 528 | 1.62 | 0.0293 | 🔴 |

**Reactome Pathways**

| pathway | description | count in network | strength | false discovery rate | |
|---------|-------------|------------------|----------|---------------------|---|
| MMU-5173214 | O-glycosylation of TSR domain-containing proteins | 3 of 37 | 2.77 | 9.55e-06 | 🔵 |

**Protein Domains (Pfam)**

| domain | description | count in network | strength | false discovery rate | |
|--------|-------------|------------------|----------|---------------------|---|
| PF01421 | Reprolysin (M12B) family zinc metalloprotease | 2 of 35 | 2.62 | 0.0189 | 🟢 |
| PF01562 | Reprolysin family propeptide | 2 of 53 | 2.44 | 0.0211 | 🟡 |

**Protein Domains and Features (InterPro)**

| domain | description | count in network | strength | false discovery rate | |
|--------|-------------|------------------|----------|---------------------|---|
| IPR041645 | ADAMTS, cysteine-rich domain 2 | 2 of 18 | 2.91 | 0.0056 | 🟣 |
| IPR010909 | PLAC | 2 of 18 | 2.91 | 0.0056 | 🟢 |
| IPR045371 | ADAMTS/ADAMTS-like, cysteine-rich domain 3 | 2 of 24 | 2.78 | 0.0058 | 🔵 |
| IPR013273 | ADAMTS/ADAMTS-like | 2 of 24 | 2.78 | 0.0058 | 🟠 |
| IPR010294 | ADAMTS/ADAMTS-like, Spacer 1 | 2 of 25 | 2.77 | 0.0058 | 🟣 |
| IPR036383 | Thrombospondin type-1 (TSP1) repeat superfamily | 3 of 65 | 2.53 | 0.00020 | 🔴 |
| IPR000884 | Thrombospondin type-1 (TSP1) repeat | 3 of 65 | 2.53 | 0.00020 | ⚪ |
| IPR002870 | Peptidase M12B, propeptide | 2 of 53 | 2.44 | 0.0164 | ⚫ |
| IPR001590 | Peptidase M12B, ADAM/reprolysin | 2 of 54 | 2.43 | 0.0164 | 🟢 |
| IPR024079 | Metallopeptidase, catalytic domain superfamily | 2 of 97 | 2.18 | 0.0429 | 🔴 |

*(less ...)*

**Protein Domains (SMART)**

| domain | description | count in network | strength | false discovery rate | |
|--------|-------------|------------------|----------|---------------------|---|
| SM00209 | Thrombospondin type 1 repeats | 3 of 65 | 2.53 | 2.33e-05 | 🔵 |

**Fig 4. Functional analysis of Adamts3, Adamts18 and Sspo proteins.** Colors indicate the type of function performed in the network. Created in STRING v. 12. Adamts3 – A disintegrin-like and metallopeptidase (reprolysin type) with thrombospondin type 1 motif, 3: Adamts 18 – A disintegrin and metalloproteinase with thrombospondin motifs 18, Sspo – SCO-spondin. Colors indicate the process related to the GO-term and biological pathways.

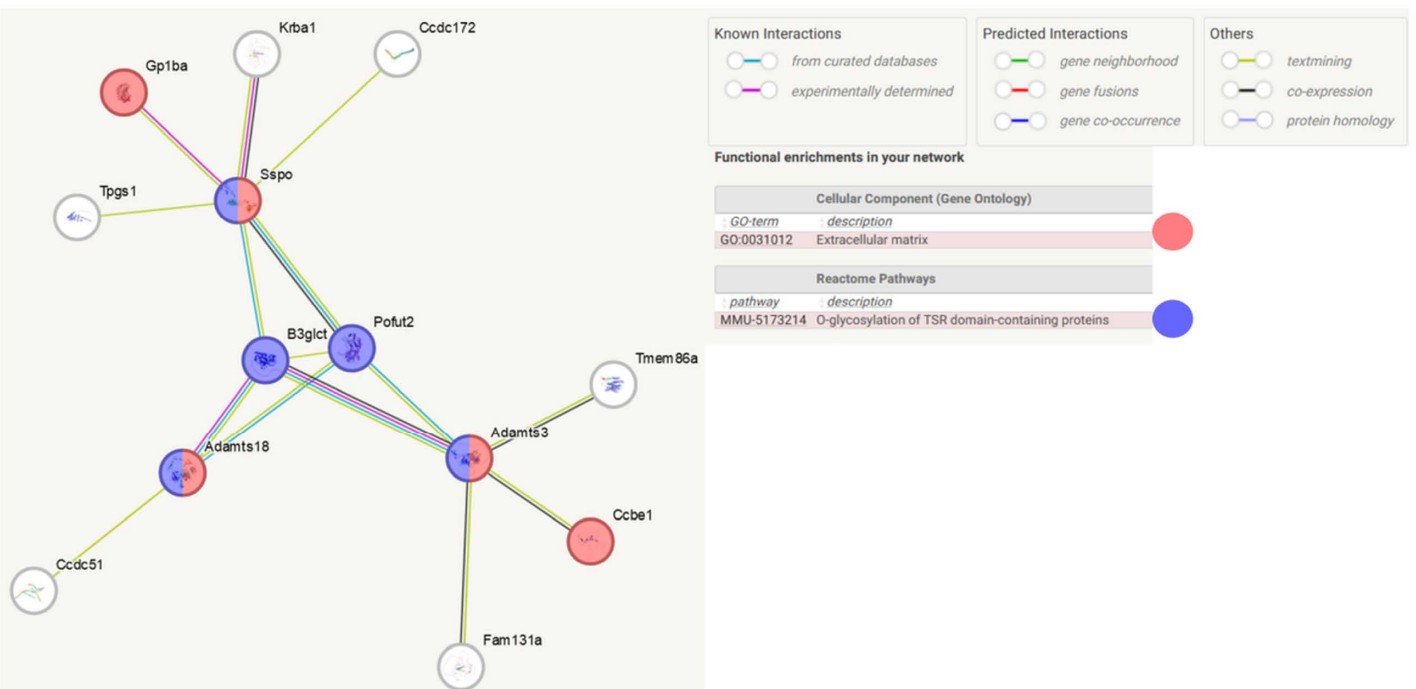

**Fig 5. Proteins involved in extracellular matrix and o-glycosylation of TSR domain-containing proteins.** Protein-protein network was created in STRING v. 12 database. Limited to *Mus Musculus*. Network nodes represent proteins. Edges represent protein-protein associations. Created in STRING v. 12. Gp1ba - Platelet glycoprotein Ib alpha chain, Krba1 – Protein KRBA1, Tmeme86a - Lysoplasmalogenase-like protein TMEM86A, Ccdc51- Mitochondrial potassium channel; Mitochondrial potassium channel located in the mitochondrial inner membrane, Fam131a - Protein FAM131A, Ccdc172 – Coiled-coil domain-containing protein 172, Tpgs1 – Tubulin polyglutamylase complex subunit 1, Ccbe1 – Collagen and calcium-binding EGF domain-containing protein 1, Pofut2 – GDP-fucose protein O-fucosyltransferase 2, B3gltc - Beta-1,3-glucosyltransferase, Adamts3 – A disintegrin-like and metallopeptidase (reprolysin type) with thrombospondin type 1 motif, 3; Adamts 18 – A disintegrin and metalloproteinase with thrombospondin motifs 18, Sspo – SCO-spondin. The color indicates the process related to the GO-term and biological pathways.

## Direct interaction with RNA

We found that RNA-RNA interactions are associated with GO-terms involved in biological process (BP), cellular component (CC) and molecular function (MF), Fig 6; S2 Table). In BP term we observed GO associated with anatomical structure development (GO:0048856), developmental process (GO:0032502), multicellular organismal process (GO:0032501), multicellular organism development (GO:0007275), cell differentiation (GO:0030154), cellular developmental process (GO:0048869), response to stress (GO:0006950), cellular response to chemical stimulus (GO:0070887), positive regulation of cellular process (GO:0048522), regulation of developmental process (GO:0050793), organonitrogen compound metabolic process (GO:1901564) and cell communication (GO:0007154; Figs 6 and 7). The largest number of genes that showed RNA-RNA interactions and associated with the BP term were involved in the multicellular organismal process (GO:0032501) and the PI3K-Akt signaling pathway (KEGG: mmu04151; Table 2, S1 Fig.).

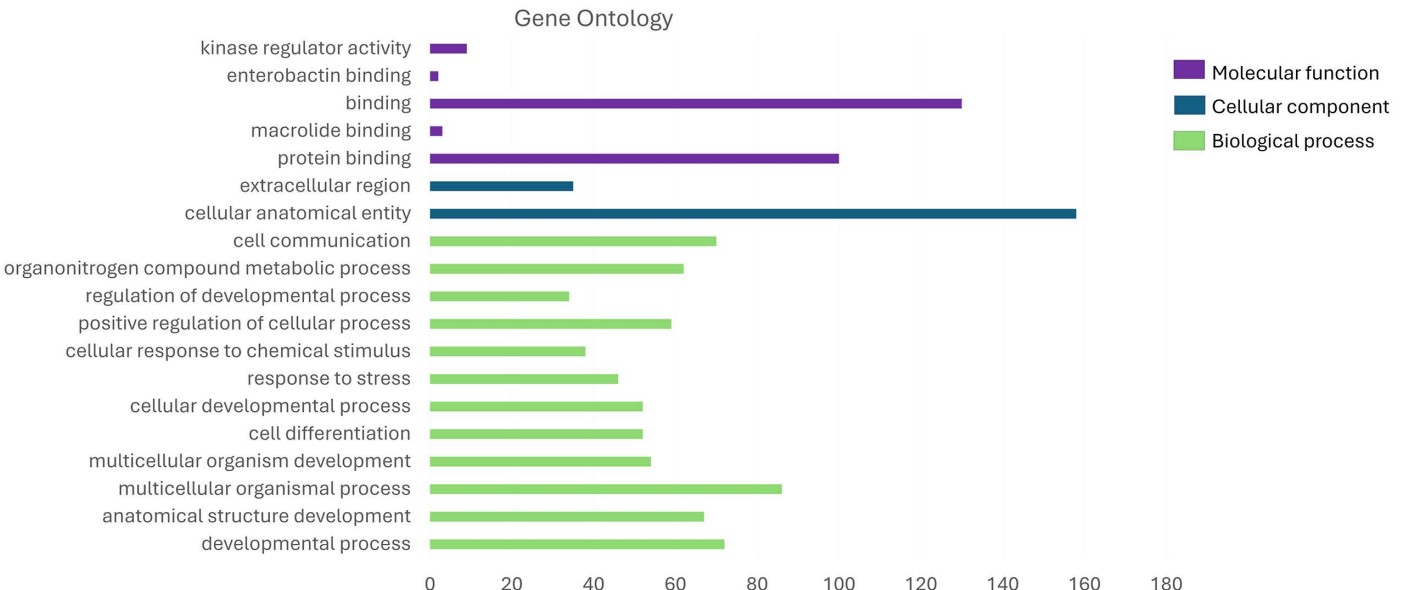

**Fig 6. Summary visualization of GO terms associated with RNA-RNA interactions.** GO terms under MF (purple), CC (dark blue) and BP (green). Cellular anatomical entity was the most enriched GO term. The X-axis indicates the number of genes involved in GO-term.

### Trans-acting

Trans-correlations were associated with five BP and two MF terms (Table 3). Among BP we distinguish anatomical structure development (GO:0048856), developmental process (GO:0032502), multicellular organism development (GO:0007275), multicellular organismal process (GO:0032501), organonitrogen compound metabolic process (GO:1901564). Moreover, we observed trans-correlations in MF associated with binding (GO:0005488) and protein binding (GO:0005515). All trans interactions are listed in S3 Table.

### Expression of Snhg15 lncRNA in the lumbar spinal cord

An elevated expression of *Snhg15* was detected in T1D spinal cord P ≤ 0.01 (Fig 8A). The expression of *Snhg15* lncRNA confirmed the results obtained from Next generation sequencing (Fig 8B). Negative controls confirmed the specificity of the primers and the carefully selected primer annealing conditions. We chose this lncRNA because of its function and interaction with other genes involved in diabetic neuropathy [3].

### Discussion

We identified 277 differentially expressed transcripts in the T1D spinal cord. The novel aspect of our study is that the two transcriptomes (i.e., determined in T1D and non-diabetic) were compared to identify the known lncRNAs that are uniquely expressed in the mouse spinal cord. Among the differentially expressed lncRNAs in the mouse spinal cord, transcripts engaging in kinase regulator activity, enterobactin binding, binding, macrolide binding, extracellular region, cellular anatomical entity, cell communication, organonitrogen compound metabolic process, regulation of developmental process, positive regulation of cellular process, cellular response to chemical stimulus, response to stress, cellular developmental process, cell differentiation, multicellular organismal process, anatomical structure development as well as developmental process were found. Nevertheless, our results revealed that lncRNAs are involved in PI3K-Akt signaling pathway. Therefore, our findings confirm the crucial role that the spinal cord may play in the progression of peripheral neuropathy in

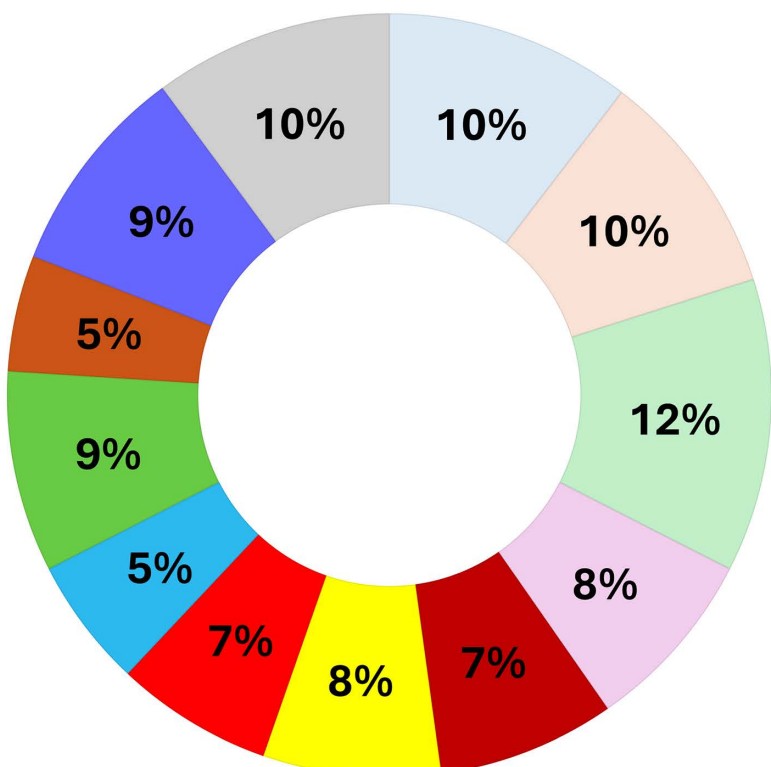

**Fig 7. GO-BP term enrichment analysis associated with RNA-RNA interactions.** Top GO terms under BP are developmental process, cell communication, anatomical structure development and multicellular organismal process. Colors indicate processes related to the GO-term.

T1D mice. We revealed that not only peripheral nervous system, but also central nervous system may be affected in the progression of diabetic peripheral neuropathy [2–3].

Spinal cord role in the progression of diabetic peripheral neuropathy has been overlooked for many decades [3,7]. The first indications of pathological change in the spinal cord during diabetic neuropathy were reported in 1960s [10–11]. Our previous results revealed that *cathepsin E* (*CTSE*) is expressed in the spinal cord and sciatic nerve in both T1D and T2D [2] and itis up regulated in a damaged spinal cord [2]. Therefore, it is plausible to speculate that the onset of neurodegenerative diseases may have its origin in the spinal cord. Our results confirm that spinal cord plays a role in the progression of diabetic peripheral neuropathy in mice.

LncRNAs may regulate expression of genes through epigenetic mechanisms [29]. The epigenome is sensitive to microenvironmental changes. Long-term hyperglycemia induces alternations in tissues and cells [30–31]. Hence, T1D may

**Table 2. The most enriched biological pathway. Genes that showed RNA-RNA interactions.**

**PI3K-Akt signaling pathway (KEGG: mmu04151)**

| Ensembl ID | Official Symbol | Description |
|---|---|---|
| ENSMUST00000177197 | *Chrm1* | cholinergic receptor, muscarinic 1 |
| ENSMUST00000125346 | *Pkn3* | protein kinase N3 |
| ENSMUST00000107571 | *Lpar1* | lysophosphatidic acid receptor 1 |
| ENSMUST00000124203 | *Sgk1* | serum/glucocorticoid regulated kinase 1 |
| ENSMUST00000020308 | *Ddit4* | DNA-damage-inducible transcript 4 |
| ENSMUST00000171265 | *Sgk3* | serum/glucocorticoid regulated kinase 3 |
| ENSMUST00000023829 | *Cdkn1a* | cyclin dependent kinase inhibitor 1A |
| ENSMUST00000199615 | *Egf* | epidermal growth factor |
| ENSMUST00000039164 | *Lpar3* | lysophosphatidic acid receptor 3 |
| ENSMUST00000029547 | *Creb3l4* | cAMP responsive element binding protein 3-like 4 |

**Table 3. Differentially expressed genes associated with BP and MF terms. Trans-acting genes.**

| Category name, * Subcategory name | Genes, Official symbol | Number |
|---|---|---|
| **Biological process** | | |
| GO:0032501 multicellular organismal process | *Rab7b, Myoc, Sgk1, Enpp1, Ddit4, Aire, Icos, Myola, Pla2g3, Ddc, Gjc2, Per1, Tbx2, Mfsd2b, Nfkbia, Nkx2–9, Ucn3, Akr1c14, Serpinb1a, Shld3, Setdb2, Maff, Acr, Serpind1, Dusp1, Spdef, Cdkn1a, Nrtn, Cabyr, Il33, Nrap, Fut7, Ptgds, Ada, Dcst2, Creb3l4, Oaz3, Ctsk, Ugt8a, Egf, Rrh, Slc9b2, Ccn1, Lpar3, Gabrr2, Lpar1, Mfsd2a, Crybg2, Asic3, Adamts3, Sh2b2, Clcn1, Sspo, Nat8f6, Hif3a, Nanos2, Meiosin, Klc3, Zfp36, Alpk3, Insc, Trim72, Itgad, Mki67, Ascl2, Hmgb2, Il12rb1, Asf1b, Adamts18, Or8b53* | 70 |
| GO:0032502 developmental process | *Rab7b, Myoc, Sgk1, Enpp1, Ddit4, Aire, Icos. Pla2g3, Ddc, Gjc2, Tbx2, Nfkbia, Nkx2–9, Akr1c14, Shld3, Setdb2, Maff, Ankrd33, Dusp1, Spdef, Cdkn1a, Nrtn, Cabyr, Cdc42ep2, Il33, Nrap, Fut7, Spdef, Ada, Creb3l4, Oaz3, Ctsk, Ugt8a, Egf, Slc9b2, Ccn1, Lpar3, Lpar1, Mfsd2a, Crybg2, Adamts3, Sh2b2, Sspo, Nat8f6, Ninj2, Hif3a, Nanos2, Meiosin, Klc3, Zfp36, Alpk3, Imsc, Trim72, Mki67, Ascl2, Tgfbr3l, Hmgb2, Asf1b, Ccl17, Adamts18* | 60 |
| GO:0048856 anatomical structure development | *Rab7b, Myoc, Sgk1, Ddit4, Aire, Icos, Pla2g3, Ddc, Gjc2, Tbx2, Nfkbia, Nkx2–9, Shld3, Akr1c14, Setdb2, Maff, Ankrd33, Dusp1, Spdef, Cdkn1a, Nrtn, Cabyr, Cdc42ep2, Il33, Nrap, Fut7, Ctsk, Lpar1, Lpar3, Slc9b2, Ccn1, Ada, Ugt8a, Egf* | 57 |
| * GO:0007275 multicellular organism development | *Rab7b, Myoc, Sgk1, Ddit4, Aire, Pla2g3, Ddc, Gjc2, Tbx2, Nfkbia, Nkx2–9, Akr1c14, Shld3, Setdb2, Maff, Dusp1, Spdef, Cdkn1a, Nrtn, Il33, Nrap, Fut7, Ada, Ctsk, Ugt8a, Egf, Slc9b2, Ccn1, Lpar3, Lpar1, Mfsd2a, Crybg2, Adamts3, Sh2b2, Sspo, Nat8f6, Hif3a, Zfp36, Alpk3, Imsc, Mki67, Ascl2, Hmgb2, Asf1b, Adamts18* | 46 |
| GO:1901564 organonitrogen compound metabolic process | *Sgk1, Enpp1, Ddit4, Aire, Pla2g3, Ddc, Gjc2, Per1, Nfkbia, Serpinb1a, Cdc14b, Galnt15, Cideb, Setdb2, Hr, Acr, Serpind1, Dusp1, Fkbp5, Cdkn1a, Gnmt, Spink10, Il33, Fut7, Pla2g4e, Trib3, Ada, Oaz3, Ctsk, Ovgp1, Ugt8a, Egf, Ccn1, Mob3b, Mfsd2a, Map3k6, Adamats3, Tfr2, Nat8f7, Nat8f6, Asprv1, Gxylt2, Nanos2, Hipk4, Zfp36, Ttll13, Alpk3, Kctd21, Acsm3, Trim72, Il12rb1, Fbxl9, Adamts18, mt-Ti* | 54 |
| **Molecular function** | | |
| GO:0005488 binding | *Rab7b, Myoc, Dnah14, Sgk1, Enpp1, Ddit4, Aire, Icosl, Myo1a, Castor1, Ddc, Per1, Tbx2, Arl4d, 1810010H24Rik, Nfkbia, Nkx2–9, Ucn3, Akr1c14, Pla2g3, Serpinb1a, Galnt15, Cideb, Setdb2, Hr, Maff, Acr, Ankrd33, Serpind1, Dynlt2a3, Dusp1, Spdef, Fkbp5, Cdkn1a, Myo1f, Ly6g6e, Ly6g5b, Gnmt, Plin5, Nrtn, Slc3a1, Cabyr, Spin10, Cdc42ep2, Il33, Nrap, Mcm10, Yme1, Ptgds, Phyhd1, Pla2g4e, Trib3, Ada, Phactr3, Fcrl, Creb3l4, Oaz3, Ctsk, Inka2, Ovgp1, Egf, Etnppl, Slc9b2, Ccn1, Lpar3, Gabrr2, Mob3b, Lpar1, Map3k6, Crybg2, 1700109H08Rik, Adamts3, Sh2b2, Tfr2, Rasl11a, Clcn1, Sspo, Zfand4, Hifa3, Nanos2, Meiosin, Klc3, Hipk4, Zfp36, Ttll13, Alpk3, Kctd21, Insc, Acsm3, Trim72, Itgad, Mki67, Pnpla2, Tspan4, Ascl2, Tgfbr3l, Hmgb2, Tma16, Il12rb1, Asf1b, Mt2, Mt1, Ccl17, Fbxl9, Adamats18, Or8b53, mt-Ti, Hsf3, Pla2g3* | 109 |
| GO:0005515 protein binding | *Myoc, Dnah14, Sgk1, Enpp1, Ddit4, Aire, Icosl, Myo1a, Castor1, Ddc, Per1, Tbx2, 1810010H24Rik, Nfkbia, Ucn3, Pla2g3, Serpinb1a, Cideb, Setdb2, Hr, Acr, Ankrd33, Dynlt2a3, Dusp1, Fkbp5, Cdkn1a, Myo1f, Ly6g6e, Ly6g5b, Gnmt, Plin5, Nrtn, Slc3a1, Cabyr, Spin10, Cdc42ep2, Il33, Nrap, Mcm10, Yme1, Trib3, Phactr3, Fcrl, Oaz3, Ctsk, Inka2, Egf, Slc9b2, Ccn1, Lpar3, Gabrr2, Lpar1, Sh2b2, Tfr2, Rasl11a, Clcn1, Zfand4, Sspo, Hifa3, Nanos2, Meiosin, Klc3, Zfp36, Ttll13, Kctd21, Insc, Trim72, Itgad, Mki67, Pnpla2, Tspan4, Ascl2, Tgfbr3l, Hmgb2, Il12rb1, Asf1b, Ccl17, Fbxl9* | 78 |

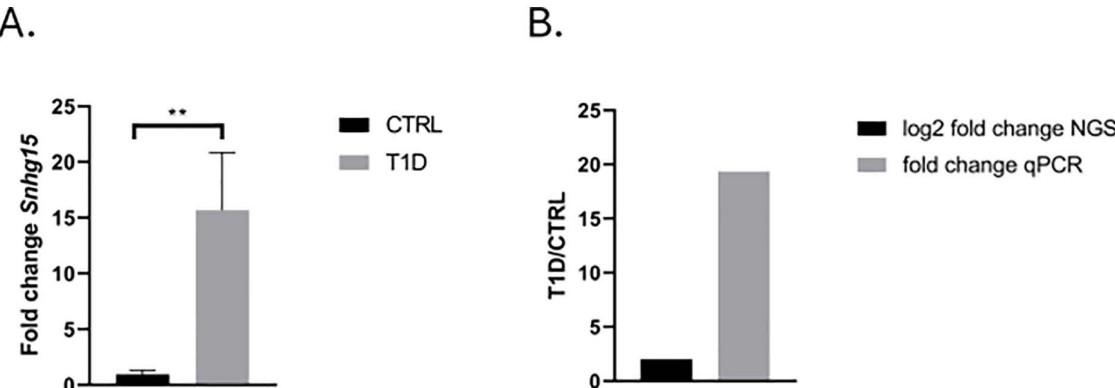

**Fig 8. Results of Next Generation Sequencing validation with qPCR. A.** Expression of Snhg15 lncRNA in the lumbar spinal cord. **B.** Fold change of *Snhg15* determined by Next Generation Sequencing was confirmed by qPCR. T1D – type 1 diabetes, CTRL – control.

contribute to alternations in the expression of lncRNAs. Malfunctions in the expression of lncRNAs and thus in the epigenetic factors may induce diabetic perturbations [32]. Studies indicated that lncRNAs are present in central nervous system during the course of neurodegenerative diseases, suggesting that malfunctions in the expression of these molecules may cause perturbations in neuronal cells [33–34].

We found that in prolonged hyperglycemia the expression of *Snhg15* lncRNA was elevated in the spinal cord of T1D mice. The analysis indicated that *Snhg15* belongs to the group of lncRNAs which participate in the modulation of vascular endothelial cell function. Previous studies have shown that the expression of *Snhg15* decreases in the hind limbs of mice with diabetes [35]. Nevertheless, we observed that the expression of *Snhg15* was elevated in the spinal cord of T1D mice.

Numerous studies indicated that long-term diabetes leads to disturbances in the vascular system [36–37]. This phenomenon may suggest that *Snhg15* may promote angiogenesis in the spinal cord harvested from T1D mice. Angiogenesis has pathogenic importance during progression of retinopathy [38]. Retinopathy is a common pathology in diabetic patients [39]. Spinal cord injury may initiate the process of angiogenesis [40]. However, angiogenic processes may not be sufficient for the regeneration of axons in the spinal cord during diabetes [41–43]. Our previous studies showed the elevated expression of *thioredoxin-interacting protein* (*TXNIP*) in the T1D spinal cord of mouse [2,3]. Increased *TXNIP* expression in mouse spinal cord may induce endothelial cell dysfunction in the case of long-term hyperglycemia [42]. Nevertheless, studies indicated molecular crosstalk between *TXNIP* and *Snhg15* during diabetes [36–37]. Overall, *Snhg15* overexpression may improve endothelial function during diabetes and thus erase the negative effect of *TXNIP* in the spinal cord of mice.

However, during long-term diabetes, the amount of advanced glycation end-products (AGEs) increased [1,42]. AGEs accumulation may trigger endothelial dysfunction in spinal cord by enhancing the production of proinflammatory cytokines and perturbations in actin cytoskeleton dynamics of nervous cells and vessels in the spinal cord [2]. In serum, plasma and tissues of patients with diabetes we observe elevated level of AGEs [1]. Moreover, AGEs are the best-known ligand for RAGE [1,6]. RAGE signaling pathways are active during renal microvascular complications in diabetic patients [42]. The authors speculated that AGEs may trigger the overexpression of *Snhg15* lncRNA in the spinal cord of T1D mice. Our findings showed that *Snhg15* may be a novel biomarker of spinal cord dysfunctions during diabetes. Moreover, our data suggests that *Snhg15* may be involved in RAGE signaling pathways underlying the endothelial dysfunction in the spinal cord of T1D mice.

Evidence indicated that in endothelial cells PI3K-Akt signaling pathway plays a role in angiogenesis [44]. However, the molecular relationship between *Snhg15* and PI3K-Akt signaling pathway is still unknown. The PI3K-Akt signaling pathway plays an essential role in diabetes [3,12]. This pathway is responsible for insulin signaling and regulation of glucose metabolism [45]. However, it should be noted that the PI3K-Akt signaling pathway plays also a significant role in another

process related to diabetes disorders [12,46,47]. Evidence suggests that the PI3K-Akt signaling pathway participates in central nervous system perturbations [45]. This pathway regulates neuron growth, survival, differentiation as well as regeneration [46–47]. Evidence indicated that nerve growth factors and some neurotrophic factors are transported in neurons *via* the PI3K-Akt signaling pathway [48]. Moreover, investigations have shown that the PI3K-Akt signaling pathway may be regulated by lncRNAs [45,48]. However, further studies are needed to clarify this phenomenon.

Our results confirm that lncRNAs play a crucial role in diabetes. Nevertheless, we indicated that *Snhg15* lncRNA may be essential in spinal cord dysfunction in T1D [49]. Our *in-silico* findings also revealed that the PI3K-Akt signaling pathway may be regulated by lncRNAs. The cause of changes in the spinal cord during diabetes may be due to disruptions of endothelial cells.

We confirmed that a long-term T1D may alter the expression of lncRNAs in mouse spinal cord. Nevertheless, the role of lncRNAs as biomarkers of T1D perturbations is not established [50]. The lncRNAs may be involved in spinal cord dysfunctions during T1D. Our data indicated that lncRNAs may be responsible for dysfunctions in endothelial cells of mouse spinal cord. Therefore, damage to the vascular epithelium in the spinal cord may cause damage to neurons and, consequently, lead to diabetic neuropathy. We assume that lncRNAs may be responsible for malfunctions in nervous system during T1D. However, a better understanding of the molecular mechanisms of lncRNAs interactions with processes occurring in the nervous and vessel systems is needed [51]. Treatment of diabetic neuropathy should be directed at processes that are often not directly related to glucose regulation, such as lncRNAs control [52–55].

## Conclusion

Our data demonstrated that lncRNAs are expressed in the spinal cord harvested from T1D mice. We observed alternations in molecular pattern of spinal cord during prolonged hyperglycemia in mice. We confirmed the T1D may affect lncRNA expression in the lumbar spinal cord. Moreover, T1D elevated the expression of *Snhg15* in T1D spinal cord as well as the PI3K-Akt signaling pathway. Our research confirmed that the neglected lncRNA, *Snhg15* may be an important marker in the progression of T1D. The presented results are a continuation of previous studies on the role of the spinal cord in diabetic neuropathy. However, our results need further study and validation.

## Supporting information

**S1 Table. A table containing a complete list of lncRNAs that were differentially expressed in the T1D lumbar spinal cord in relation to control.**
(DOCX)

**S2 Table. We found that RNA-RNA interactions are associated with GO-terms involved in biological process (BP), cellular component (CC) and molecular function (MF).**
(DOCX)

**S3 Table. Trans-acting.**
(DOCX)

**S1 Fig. The most enriched biological pathway. PI3K-Akt signaling pathway (KEGG: mmu04151).** Blue rectangles without background indicate the site of lncRNA interaction.
(TIF)

## Acknowledgments

Authors would like to thank Staff at the Regenerative Medicine Laboratory and Laboratory of Stem Cells Research, University of Warmia and Mazury in Olsztyn, Warszawska 30, 10–082 Olsztyn for letting us use the laboratory space and equipment.

## Author contributions

**Conceptualization:** Kamila Zglejc-Waszak.

**Data curation:** Kamila Zglejc-Waszak, Jan Pawel Jastrzebski, Judyta Karolina Juranek.

**Formal analysis:** Kamila Zglejc-Waszak, Jan Pawel Jastrzebski.

**Funding acquisition:** Judyta Karolina Juranek.

**Investigation:** Kamila Zglejc-Waszak, Jan Pawel Jastrzebski.

**Methodology:** Kamila Zglejc-Waszak, Jan Pawel Jastrzebski.

**Project administration:** Jan Pawel Jastrzebski, Joanna Wojtkiewicz, Judyta Karolina Juranek.

**Resources:** Kamila Zglejc-Waszak, Jan Pawel Jastrzebski.

**Software:** Kamila Zglejc-Waszak, Jan Pawel Jastrzebski.

**Supervision:** Kamila Zglejc-Waszak, Jan Pawel Jastrzebski, Zenon Pidsudko, Judyta Karolina Juranek.

**Validation:** Kamila Zglejc-Waszak, Jan Pawel Jastrzebski.

**Visualization:** Kamila Zglejc-Waszak, Jan Pawel Jastrzebski, Joanna Wojtkiewicz, Zenon Pidsudko, Judyta Karolina Juranek.

**Writing – original draft:** Kamila Zglejc-Waszak, Jan Pawel Jastrzebski.

**Writing – review & editing:** Jan Pawel Jastrzebski, Joanna Wojtkiewicz, Zenon Pidsudko, Judyta Karolina Juranek.

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
