## [Decision Letter · Decision Letter 0]

5 Aug 2025

Dear Dr. Zglejc-Waszak,

Thank you for submitting your manuscript to PLOS ONE. After careful consideration, we feel that it has merit but does not fully meet PLOS ONE’s publication criteria as it currently stands. Therefore, we invite you to submit a revised version of the manuscript that addresses the points raised during the review process.

We look forward to receiving your revised manuscript.

Kind regards,

Li Shen

Academic Editor

PLOS ONE

Journal Requirements:

The authors declare no conflict of interest.

5. Please amend the manuscript submission data (via Edit Submission) to include author Judyta Juranek

6. Please amend your authorship list in your manuscript file to include author Judyta Karolina Juranek

Additional Editor Comments:

The manuscript presents a relevant and timely investigation into the role of long non-coding RNAs (lncRNAs) in diabetic neuropathy, focusing on spinal cord changes in a mouse model of type 1 diabetes. The study addresses an underexplored area and includes both bioinformatics analysis and experimental validation. However, all reviewers identified several critical issues that must be addressed before the manuscript can be considered for publication. These include insufficient contextualization of the findings within existing literature, limited validation of identified lncRNAs (with only Snhg15 experimentally confirmed), and a lack of mechanistic support for key biological claims. Moreover, the diabetic neuropathy model is not adequately described, and essential information regarding disease onset, neuropathy development, and animal selection for transcriptomic profiling is missing. Reviewers also noted concerns with figure clarity, incomplete supplementary materials, and overinterpretation of results in the Discussion. Substantial revisions are required to improve the manuscript's clarity, experimental justification, and scientific rigor.

Reviewers' comments:

Reviewer's Responses to Questions

**Comments to the Author**

1. Is the manuscript technically sound, and do the data support the conclusions?

Reviewer #1: Yes

Reviewer #2: Yes

Reviewer #3: Partly

2. Has the statistical analysis been performed appropriately and rigorously?

Reviewer #1: Yes

Reviewer #2: Yes

Reviewer #3: Yes

3. Have the authors made all data underlying the findings in their manuscript fully available?

Reviewer #1: Yes

Reviewer #2: Yes

Reviewer #3: Yes

4. Is the manuscript presented in an intelligible fashion and written in standard English?

Reviewer #1: Yes

Reviewer #2: Yes

Reviewer #3: Yes

Reviewer #1: Some remarks to authors:

General Assessment: This manuscript investigates the impact of prolonged hyperglycemia on long non-coding RNA expression in the spinal cord of diabetic mice. The study addresses an underexplored aspect of diabetic neuropathy, particularly focusing on central nervous system involvement. The bioinformatics pipeline and experimental validation of Snhg15 expression are strengths. However, several issues should be addressed before the manuscript is considered for publication.

Major Comments:

Novelty and Contextualization: The manuscript provides new data on lncRNAs in the diabetic spinal cord. However, further contextualization with recent high-throughput studies on lncRNAs in neurodegeneration and diabetes would strengthen the Introduction and Discussion.

Validation: Only Snhg15 was validated by qPCR. Please justify why no additional lncRNAs were selected for validation. Were any protein-level experiments (e.g., Western blot, immunohistochemistry) considered to confirm PI3K-Akt pathway activation?

Mechanistic Insight: The link between Snhg15 and endothelial dysfunction or RAGE signaling is hypothesized but not mechanistically demonstrated. Authors should moderate the claims or provide additional experimental support.

Figure Quality and Supplementary Data: Some figures (e.g., interaction networks) lack clarity and resolution. The supplementary tables referenced in the text (e.g., S1 Table, S2 Table) are not included in this file. These should be provided for review.

Minor Comments:

Please revise the text for grammatical accuracy and clarity. There are frequent awkward phrasings (e.g., "We confirmed pathological effect of T1D in lumbar spinal cord of the expression of lncRNAs").

Consider removing repetitive statements, especially in the Discussion.

Ensure that all GO terms and KEGG pathways are described with their biological significance.

Best regards

Reviewer #2: First of all, I would like to thank Plos One for inviting me to review the article Novel insights into neuropathy: the impact of prolonged hyperglycemia on long non-coding RNA expression.

After reading it, I can confirm that the article is pertinent and has logical conclusions.

I believe that the article is ready for release.

Thank you,

Dr João Paulo Barile

Neurologist at the Department of Neuromuscular Diseases at the Federal University of São Paulo (UNIFESP)

Reviewer #3: The analysis of long non coding RNAs is not so deeply studied as other non conding RNAs, so a study focused on lncRNAs in neuropathy is interesting. Major points should be implemented:

- the model of diabetic neuropathy is not described, information is demanded to references 2 and 18. Ref 2 is a review, ref 18 does not seem related to diabetes model. Please clarify and clear detail the model used.

- Please report data regarding the development of diabetes and neuropathy in the animals used for transcriptomic analysis; at least at the time point considered for sacrifice (6 months)

- Depending on the model, not all the animals with diabetes develop neuropathy, please report here the percentage. Were the non neuropathic animals excluded by the omic analysis?

- Snhg15 lncRNA expression was validated by RT-PCR. Why the attention was focused on this particular target? Since among several others one only was verified, the choice must be well supported.

- The discussion is mainly based on the description of Snhg15 and its relationship with PI3K (not measured here); the authors state "We assume that Snhg15 lncRNA may be responsible for malfunctions in nervous system during T1D", the authors should better analyze this hypothesis or strongly revise the discussion reducing the impact of only probabilistic suggestions

**Do you want your identity to be public for this peer review?** For information about this choice, including consent withdrawal, please see our Privacy Policy

Reviewer #1: No

Reviewer #2: **Yes: ** João Paulo Barile

Reviewer #3: No

---

## [Author Response · Author response to Decision Letter 1]

3 Sep 2025

Additional Editor Comments:

The manuscript presents a relevant and timely investigation into the role of long non-coding RNAs (lncRNAs) in diabetic neuropathy, focusing on spinal cord changes in a mouse model of type 1 diabetes. The study addresses an underexplored area and includes both bioinformatics analysis and experimental validation. However, all reviewers identified several critical issues that must be addressed before the manuscript can be considered for publication. These include insufficient contextualization of the findings within existing literature, limited validation of identified lncRNAs (with only Snhg15 experimentally confirmed), and a lack of mechanistic support for key biological claims. Moreover, the diabetic neuropathy model is not adequately described, and essential information regarding disease onset, neuropathy development, and animal selection for transcriptomic profiling is missing. Reviewers also noted concerns with figure clarity, incomplete supplementary materials, and overinterpretation of results in the Discussion. Substantial revisions are required to improve the manuscript's clarity, experimental justification, and scientific rigor.

Response: We would like to thank you for the very thorough reading of our manuscript.

Diabetic neuropathy is a vast topic with new information coming out almost every week. In depth coverage of this topic may be voluminous. We have tried here to limit the scope of our MS to aberrant lncRNAs mediated signaling in diabetic neuropathy/axonopathy and how new high throughput data may help us formulate new hypothesis. We believe that understanding this pathway may uncover targets for safe and efficacious therapeutic intervention for diabetic symmetrical axonal neuropathy treatment. We believe that the scope limitation has allowed us to better focus on how interaction between lncRNAs may play a pivotal role in diabetic neurological complications in animal models and thus human patients and, whether, the understanding of this pathway can uncover targets for safe and efficacious therapeutic intervention for diabetic symmetrical axonal neuropathy. We also do not delve into the topic of lncRNAs pathway outside the context of diabetic symmetrical axonal neuropathy and spinal cord. Neuronal injury in diabetic peripheral neuropathy has been well recognized within the peripheral nervous system for over a century. Studies in the 1960’s identified pathological alterations in spinal cord and brain structures in patients with diabetes. However, for many years after these findings, the involvement of central nervous system in diabetic peripheral neuropathy was largely overlooked until the advent of advanced neuroimaging techniques in the latter part of the 20th century. To date however no study has used high throughput RNA sequencing experiments on spinal cord in animal models of diabetic peripheral neuropathy. Thus, we aimed to find out whether there was an altered expression of lncRNAs as well as biological pathways in type 1 diabetic mice with long-term diabetic peripheral neuropathy (6 months duration of the disease).

All tissues were sampled during our previous study, therefore we limited the number of animals used in accordance with the 3R principle. Thanks to this approach, we have minimized animal suffering and the number of animals.

We have made every effort to improve our manuscript according to your suggestions and those of the reviewers.

Reviewer #1: Some remarks to authors:

General Assessment: This manuscript investigates the impact of prolonged hyperglycemia on long non-coding RNA expression in the spinal cord of diabetic mice. The study addresses an underexplored aspect of diabetic neuropathy, particularly focusing on central nervous system involvement. The bioinformatics pipeline and experimental validation of Snhg15 expression are strengths. However, several issues should be addressed before the manuscript is considered for publication.

Response: We would like to thank you for the very thorough reading of our manuscript. We are very excited by the positive critiques of our manuscript. Please find our point-by-point response below.

Major Comments:

Reviewer #1: Novelty and Contextualization: The manuscript provides new data on lncRNAs in the diabetic spinal cord. However, further contextualization with recent high-throughput studies on lncRNAs in neurodegeneration and diabetes would strengthen the Introduction and Discussion.

Response: Diabetic neuropathy is a vast topic with new information coming out almost every week. In depth coverage of this topic may be voluminous. We have tried here to limit the scope of our MS to aberrant lncRNAs mediated signaling in diabetic neuropathy/axonopathy and how new high throughput data may help us formulate new hypothesis. We believe that understanding this pathway may uncover targets for safe and efficacious therapeutic intervention for diabetic symmetrical axonal neuropathy treatment. We believe that the scope limitation has allowed us to better focus on how interaction between lncRNAs may play a pivotal role in diabetic neurological complications in animal models and thus human patients and, whether the understanding of this pathway can uncover targets for safe and efficacious therapeutic intervention for diabetic symmetrical axonal neuropathy. We also do not delve into the topic of lncRNAs pathway outside the context of diabetic symmetrical axonal neuropathy and spinal cord. Neuronal injury in diabetic peripheral neuropathy has been well recognized within the peripheral nervous system for over a century. Studies in the 1960’s identified pathological alterations in spinal cord and brain structures in patients with diabetes. However, for many years after these findings, the involvement of central nervous system in diabetic peripheral neuropathy was largely overlooked until the advent of advanced neuroimaging techniques in the latter part of the 20th century. To date however, no study has used high throughput RNA sequencing experiments on spinal cord in animal models of diabetic peripheral neuropathy. Thus, we aimed to find out whether there was an altered expression of lncRNAs as well as biological pathways in type 1 diabetic mice with long-term diabetic peripheral neuropathy (6 months duration of the disease).

However, we have added new references.

Reviewer #1: Validation: Only Snhg15 was validated by qPCR. Please justify why no additional lncRNAs were selected for validation. Were any protein-level experiments (e.g., Western blot, immunohistochemistry) considered to confirm PI3K-Akt pathway activation?

Response: Our current results are a continuation of our previous studies (Zglejc-Waszak et al. 2023).

Our current results suggest that the expression of SNHG15 lncRNAs was up-regulated in spinal cord of diabetic mice. SNHG15 lncRNA alters the expression level of target proteins like TXNIP. Increase in expression of TXNIP was reported in the plasma of diabetic patients. Dunn and co-workers (2014) showed that elevated expression of TXNIP protein may trigger endothelial dysfunctions by inhibiting synthesis of vascular endothelial growth factor (VEGF). Our study demonstrates simultaneous overexpression of SNHG15 as well as TXNIP in lumbar spinal cord of type 1 diabetic mice. We speculate that elevated expression of SNHG15 may be compensatory to overexpression of TXNIP in lumbar spinal cord in hyperglycemia. However, further studies are needed to elucidate SNHG15 lncRNA effect on the TXNIP expression pattern in diabetes.

Our previous sequencing analysis demonstrated that the most enriched categories were those related to signal transduction, with PI3K-Akt signaling pathway (mmu04151) being the most enriched pathway (Zglejc-Waszak et al. 2023). PI3K-Akt signaling pathway is engaged in multiple functions in cells, such as metabolism, cell survival, proliferation and angiogenesis in response to extracellular factors. It is also involved in the regulation of glucose level in cells and regeneration of peripheral nervous system as well as nerves growth in central nervous system. Moreover, AGE-RAGE interaction activates the PI3K-Akt pathway. Our data confirmed elevated level of RAGE protein and mRNA during type 1 diabetes in nervous system (spinal cord as well as sciatic nerve).

Fig. The expression of AGER (gene encoding RAGE) – B in the type 1 diabetic spinal cord.

Fig. The amount of RAGE protein in type 1 diabetic sciatic nerve.

Reviewer #1: Mechanistic Insight: The link between Snhg15 and endothelial dysfunction or RAGE signaling is hypothesized but not mechanistically demonstrated. Authors should moderate the claims or provide additional experimental support.

Response: We responded to this comment above. Moreover, Juranek et al. revealed that RAGE may play a key biological role in diabetic microvascular complications. Our conclusions are based on our extensive research. I have added a new reference to the current text to ensure that these conclusions are supported by scientific evidence. Nevertheless, we agree that further studies are needed to elucidate SNHG15 lncRNA role in endothelial dysfunction during type 1 diabetes. Moreover, our previous data suggests that molecular changes in spinal cord may act synergistically with RAGE signaling pathway in the peripheral nerve. In the reviewed version of the manuscript, we will try to provide arguments to support our claims.

Reviewer #1: Figure Quality and Supplementary Data: Some figures (e.g., interaction networks) lack clarity and resolution. The supplementary tables referenced in the text (e.g., S1 Table, S2 Table) are not included in this file. These should be provided for review.

Response: Thank you for your comments. We have provided for review as per suggestion. Supplementary files have been added. We have noticed that in the pdf file the figures have low resolution, but after downloading the figures (tiff) they are of very good quality.

Minor Comments:

Reviewer #1: Please revise the text for grammatical accuracy and clarity. There are frequent awkward phrasings (e.g., "We confirmed pathological effect of T1D in lumbar spinal cord of the expression of lncRNAs").

Response: Thank you for your comments. We have revised the text for grammatical accuracy and clarity. We hope that that the revised version reads clearly and accurately as per the commentary

Reviewer #1: Consider removing repetitive statements, especially in the Discussion.

Response: Thanks for your comment. We have re-edited the Discussion section.

Reviewer #1: Ensure that all GO terms and KEGG pathways are described with their biological significance.

Response: We have followed your suggestion. We hope the text of our MS is now clear.

Reviewer #2: First of all, I would like to thank Plos One for inviting me to review the article Novel insights into neuropathy: the impact of prolonged hyperglycemia on long non-coding RNA expression.

After reading it, I can confirm that the article is pertinent and has logical conclusions.

I believe that the article is ready for release.

Response: We would like to thank you for the very thorough reading of our manuscript. We are very excited by the positive critiques of our manuscript.

Reviewer #3: The analysis of long non coding RNAs is not so deeply studied as other non conding RNAs, so a study focused on lncRNAs in neuropathy is interesting. Major points should be implemented:

Response: We would like to thank you for the very thorough reading of our manuscript. We are very excited by the positive critiques of our manuscript. Please find our point-by-point response below.

Reviewer #3:

- the model of diabetic neuropathy is not described, information is demanded to references 2 and 18. Ref 2 is a review, ref 18 does not seem related to diabetes model. Please clarify and clear detail the model used.

Response: Thank you for your suggestion. We have clarified this issue in the text of our revised MS (lines:58-64). We apologize for the inaccuracies.

Reviewer #3:

- Please report data regarding the development of diabetes and neuropathy in the animals used for transcriptomic analysis; at least at the time point considered for sacrifice (6 months)

Response: We have clarified the issue in the text of our revised MS (lines:64-66). We apologize for the inaccuracies.

Reviewer #3:

- Depending on the model, not all the animals with diabetes develop neuropathy, please report here the percentage. Were the non neuropathic animals excluded by the omic analysis?

Response: We have clarified this information in our revised MS (lines:64-66). We apologize for the inaccuracies. Briefly, we monitored blood glucose levels after five days after the last dose of STZ injection and for the following weeks to confirm diabetes. Nevertheless, we have many years of experience in inducing type 1 diabetes in a mouse model of the disease. Our experience has been supported by numerous scientific publications since 2013. Laboratorymouse is an excellent model for the induction of type 1 diabetes. Evidence supporting our thesis is provided by numerous publications using a mouse model in studies on the type 1 diabetes [dodać piśmiennictwo na potwierdzenie tej tezy]

We have put a lot of effort into the presented analyses and studies in our MS. Moreover, mouse experiments were performed in accordance with the Local Ethical Committee of Experiments on Animals in Olsztyn (Poland; decision no. 57/2019) and the studies were reported in accordance with ARRIVE guidelines (https://arriveguidelines.org) and the three Rs principle. The 3 Rs stand for Replacement, Reduction and Refinement. Additionally, we report annually on the number of animals used in studies. We cannot exceed the number of animals used in study that is permitted by the Local Animal Ethics Committee. Hence, the number of animals that will not develop diabetes is less than 1%.

Reviewer #3:

- Snhg15 lncRNA expression was validated by RT-PCR. Why the attention was focused on this particular target? Since among several others one only was verified, the choice must be well supported.

Response: Our previous studies showed that the elevated expression of thioredoxin-interacting protein (TXNIP) in the T1D spinal cord of mouse (Zglejc-Waszak et al. 2023). Increased TXNIP expression in mouse spinal cord may induce endothelial cell dysfunction in the case of long-term hyperglycemia. Nevertheless, studies indicated molecular crosstalk between TXNIP and Snhg15 during diabetes. Overall, Snhg15 overexpression may improve endothelial function during diabetes and thus erase the negative effect of TXNIP in the spinal cord of mice. The analysis of Sngh15 was well-thought-out and results from the analysis of our previous research data.

In our previous work we validated data obtained from diabetic lumbar spinal cord samples. Our current results are a continuation of our previous studies (Zglejc-Waszak et al. 2023).

Reviewer #3:

- The discussion is mainly based on the description of Snhg15 and its relationship with PI3K (not measured here); the authors state "We assume that Snhg15 lncRNA may be responsible for malfunctions in nervous system during T1D", the authors should better analyze this hypothesis or strongly revise the discussion reducing the impact of only probabilistic suggestions.

Response: Thank you for your comment. We have slightly edited the Discussion section to soften its tone. However, our current results are a continuation of our previous studies (Zglejc-Waszak et al. 2023). Thus, we have well-established evidence that the spinal cord plays a key role in diabetic neuropathy in a mouse model of the disease. Identifying molecular changes occurring in the spinal cord during diabetes may be a milestone in inhibiting the development of peripheral neuropathy.

---

## [Decision Letter · Decision Letter 1]

24 Sep 2025

Novel insights into neuropathy: the impact of prolonged hyperglycemia on long non-coding RNA expression

PONE-D-25-24198R1

Dear Dr. Zglejc-Waszak,

We’re pleased to inform you that your manuscript has been judged scientifically suitable for publication and will be formally accepted for publication once it meets all outstanding technical requirements.

Kind regards,

Li Shen

Academic Editor

PLOS ONE

Additional Editor Comments (optional):

Reviewers' comments:

Reviewer's Responses to Questions

**Comments to the Author**

Reviewer #2: All comments have been addressed

Reviewer #3: All comments have been addressed

2. Is the manuscript technically sound, and do the data support the conclusions?

Reviewer #2: Yes

Reviewer #3: Yes

3. Has the statistical analysis been performed appropriately and rigorously?

Reviewer #2: Yes

Reviewer #3: Yes

4. Have the authors made all data underlying the findings in their manuscript fully available?

Reviewer #2: Yes

Reviewer #3: Yes

5. Is the manuscript presented in an intelligible fashion and written in standard English?

Reviewer #2: Yes

Reviewer #3: Yes

Reviewer #2: First, I would like to thank PLOS ONE for the opportunity to review the manuscript "New Insights into Neuropathy: The Impact of Prolonged Hyperglycemia on Long Noncoding RNA Expression."

The article is coherently written, with pertinent conclusions. I confirm that I accept the publication of this article.

Thank you,

Dr. João Paulo Barile, Neurologist

Reviewer #3: The authors improved the manuscript according to suggestions, in my opinion it can be accepted by the journal

**Do you want your identity to be public for this peer review?** For information about this choice, including consent withdrawal, please see our Privacy Policy

Reviewer #2: No

Reviewer #3: No

---

## [Editor Report · Acceptance letter]

PONE-D-25-24198R1

PLOS ONE

Dear Dr. Zglejc-Waszak,

I'm pleased to inform you that your manuscript has been deemed suitable for publication in PLOS ONE. Congratulations! Your manuscript is now being handed over to our production team.

Kind regards,

on behalf of

Dr. Li Shen

Academic Editor

PLOS ONE